# CoFrGeNet: Continued Fraction Architectures for Language Generation

Amit Dhurandhar [1]    Vijil Chenthamarakshan [1]    Dennis Wei [1]    Tejaswini Pedapati [1]
Karthikeyan Natesan Ramamurthy [1]    Rahul Nair [2]

## Abstract

Transformers are arguably the preferred architecture for language generation. In this paper, inspired by continued fractions, we introduce a new function class for generative modeling. The architecture family implementing this function class is named CoFrGeNets - Continued Fraction Generative Networks. We design novel architectural components based on this function class that can replace Multi-head Attention and Feed-Forward Networks in Transformer blocks while requiring much fewer parameters. We derive custom gradient formulations to optimize the proposed components more accurately and efficiently than using standard PyTorch-based gradients. Our components are a plug-in replacement requiring little change in training or inference procedures that have already been put in place for Transformer-based models thus making our approach easy to incorporate in large industrial workflows. We experiment on two very different transformer architectures GPT2-xl (1.5B) and Llama3 (3.2B), where the former we pre-train on OpenWebText and GneissWeb, while the latter we pre-train on the docling data mix which consists of nine different datasets. Results show that the performance on downstream classification, Q& A, reasoning and text understanding tasks of our models is competitive and sometimes even superior to the original models with $\frac{2}{3}$ to $\frac{1}{2}$ the parameters and shorter pre-training time. We believe that future implementations customized to hardware will further bring out the true potential of our architectures.

## 1. Introduction

Since OpenAI's ChatGPT release at the end of 2022, Large Language Models (LLMs) (Radford et al., 2019) have been getting increasingly infused into multiple user applications and platforms across the world. The most prevalent architecture behind these models is the Transformer architecture (Vaswani et al., 2017) consisting of a (multi-head) Attention block and a Feed Forward Network (FFN) with a large hidden layer. In this paper, we propose novel architectural components based on a significantly different function class inspired by continued fractions. Taking inspiration from (Puri et al., 2021), where continued fraction architectures *CoFrNets* were introduced for the supervised setting, we build new architectures for the generative setting providing alternatives for attention and FFN in Transformer blocks.

Given a canonical form for continued fractions $a_0 + \frac{1}{a_1 + \frac{1}{a_2 + \cdots}}$ (ladder like structure) where, $a_k$s are complex numbers, CoFrNets (Puri et al., 2021) were introduced for supervised learning problems where in place of the $a_k$s, linear functions of the input $x \in \mathbb{R}^p$ are computed by taking the inner product of $x$ with weight vector $w_k \in \mathbb{R}^p$ in each layer $k$ (or also referred to as step of the ladder).[1] The reciprocal of the function thus far is applied as a nonlinearity in each layer leading to the following kind of form for a single CoFrNet ladder:

$$w_0 x + \frac{1}{w_1 x + \frac{1}{w_2 x + \cdots}} \qquad (1)$$

Here $w_k$s are the learnable parameters. Essentially, the input $x$ is passed to each layer which gets multiplied by the corresponding parameter vectors and the reciprocal of the values of the previous layer are added to this. This simple architecture was shown to have universal approximation capabilities when we ensemble enough of these ladders. However, the above contributions were for the supervised setting and it is not clear if such architectures can also be built for representation learning and sequence generation, where we: i) Need to produce multi-dimensional outputs, ii) learn richer functions and iii) model sequences causally i.e. learning parameters that depend only on prior tokens. Moreover, the $\frac{1}{x}$ non-linearity is inefficient to compute in

[1]IBM Research, Yorktown Heights, NY, USA [2]IBM Research, Dublin, Ireland. Correspondence to: Amit Dhurandhar <adhuran@us.ibm.com>.

*Proceedings of the 43$^{rd}$ International Conference on Machine Learning*, Seoul, South Korea. PMLR 306, 2026. Copyright 2026 by the author(s).

---

[1]A constant term is assumed to be absorbed in $x$.

forward and backward passes especially when the depth $d$ and number of ladders $L$ is large. This is because one has to compute the inverse $d \times L$ times and it is known that division is many times slower than multiplication in modern hardware. We address the above challenges in this paper by making the following contributions that distinguish it significantly from (Puri et al., 2021):

- We propose *novel continued fraction architectures* for (causal) attention and FFNs as depicted in Figure 1. We call our architecture with both components replaced as **Co**ntinued **Fr**action **Ge**nerative **Net**work (CoFrGeNet). We report results replacing either FFN or attention or both offering the possibility to the user of replacing only one or both of the components for their application. Even replacing one component can offer significant parameter and training time savings as seen in our experiments.

- We propose an *alternative representation* for the ladders and derive custom formulas for the gradients that reduces the number of divisions from $d$ to a constant of just $1$ for a $d$-depth ladder. This greatly enhances both training and inference efficiency.

- We propose a *custom training schedule* to update CoFr-GeNet parameters. This is described in section 5.

- We pre-train our models on OpenWebText (OWT) (Gokaslan et al., 2019), GneissWeb (Gohari et al., 2025) and the docling data mix (Team, 2024) showing that our models are *competitive or outperform* the corresponding Transformer models. We compare with Transformers since we are replacing its components making it a fair comparison. For an apples-to-apples comparison with other model architectures such as Mamba (Gu & Dao, 2024) one would want to replace its components with novel (to be designed) CoFrNet components, which would be a significant independent contribution in itself that we leave for the future.

## 2. Preliminaries

The generalized form for a continued fraction is $a_0 + \frac{b_1}{a_1 + \frac{b_2}{a_2 + \cdots}}$, where $a_k$s and $b_k$s can be complex numbers. If none of the $a_k$ or $b_k$ are zero $\forall k \in \mathbb{N}$, then using equivalence transformations (Jones & Thron, 1980), one can create simpler equivalent forms where either the $b_k = 1$ or the $a_k = 1 \ \forall k \in \mathbb{N}$, with $a_0 = 0$ in the latter form. A more concise way to write these two forms is as follows: i) $a_0 + \frac{1}{a_1 + \frac{1}{a_2 + \cdots}} \equiv a_0 + \frac{1}{a_1 +} \frac{1}{a_2 + \cdots}$ and ii) $\frac{b_1}{1 + \frac{b_2}{1 + \cdots}} \equiv \frac{b_1}{1+} \frac{b_2}{1 + \cdots}$. Form i) is known as the *canonical form*. One of the nice properties of continued fractions is that they are the best possible rational approximation to real numbers (Jones & Thron, 1980; Milton, 2011).

In this work, we consider continued fractions in canonical form, with partial numerators $b_k = 1$ for $k = 1, \ldots, d$ and depth $d$. We thus view continued fractions as functions $f$ of the partial denominators, where we separate $a_0$ from the others and use $a := (a_1, \ldots, a_d)$ as a shorthand. Hence,

$$f(a_0, a) = a_0 + \frac{1}{a_1 +} \frac{1}{a_2 +} \cdots \frac{1}{a_{d-1} +} \frac{1}{a_d} = a_0 + \tilde{f}(a), \tag{2}$$

where we also define $\tilde{f}(a)$ as the "fractional part" of $f(a_0, a)$. Another way of representing a continued fraction is in terms of *continuants*, which we describe next. The continued fraction in (2) can be expressed as the following ratio of polynomials $K_{d+1}$ and $K_d$,

$$f(a_0, a) = \frac{K_{d+1}(a_0, \ldots, a_d)}{K_d(a_1, \ldots, a_d)}. \tag{3}$$

Polynomials $K_d$, $K_{d+1}$ are part of a sequence of polynomials $K_k$, $k = 0, 1, \ldots$, known as *continuants*. They satisfy the recursion

$$K_0 = 1, \qquad K_1(a_d) = a_d, \tag{4}$$
$$K_k(a_{d-k+1}, \ldots, a_d) = a_{d-k+1} K_{k-1}(a_{d-k+2}, \ldots, a_d)$$
$$+ K_{k-2}(a_{d-k+3}, \ldots, a_d). \tag{5}$$

Using (5), (3) can also be written as $f(a_0, a) = a_0 + \frac{K_{d-1}(a_2, \ldots, a_d)}{K_d(a_1, \ldots, a_d)}$ and hence,

$$\tilde{f}(a) = \frac{K_{d-1}(a_2, \ldots, a_d)}{K_d(a_1, \ldots, a_d)}. \tag{6}$$

We will exploit the formalism of continuants later for two purposes: first, as a means of computing continued fractions, and second, to derive closed-form expressions for their gradients. This leads to benefits in the forward direction, in terms of speeding up inference, and also in the backward direction, speeding up training, compared to standard back-propagation through the multiple layers of a continued fraction. While the original CoFrNet work (Puri et al., 2021) used this formalism for the limited purpose of local feature-based explanations, here we derive new results making them an integral part in training our architectures.

To construct networks out of continued fractions, we let the partial denominators $a_k$ be affine functions of an input $x$, $a_k = w_k x$, where $w_k$ is a row vector and a $1$ is prepended to the elements of $x$ so that the corresponding coefficient $w_{k0}$ is the intercept or "bias" term. We will often refer to a continued fraction with $a_k = w_k x$ as a (CoFrNet) "ladder", and we will also construct ensembles of such ladders. Throughout the paper we denote the input or embedding dimension by $p$, the number of ladders in an ensemble by $L$, and sequence length by $l$, unless specified otherwise.

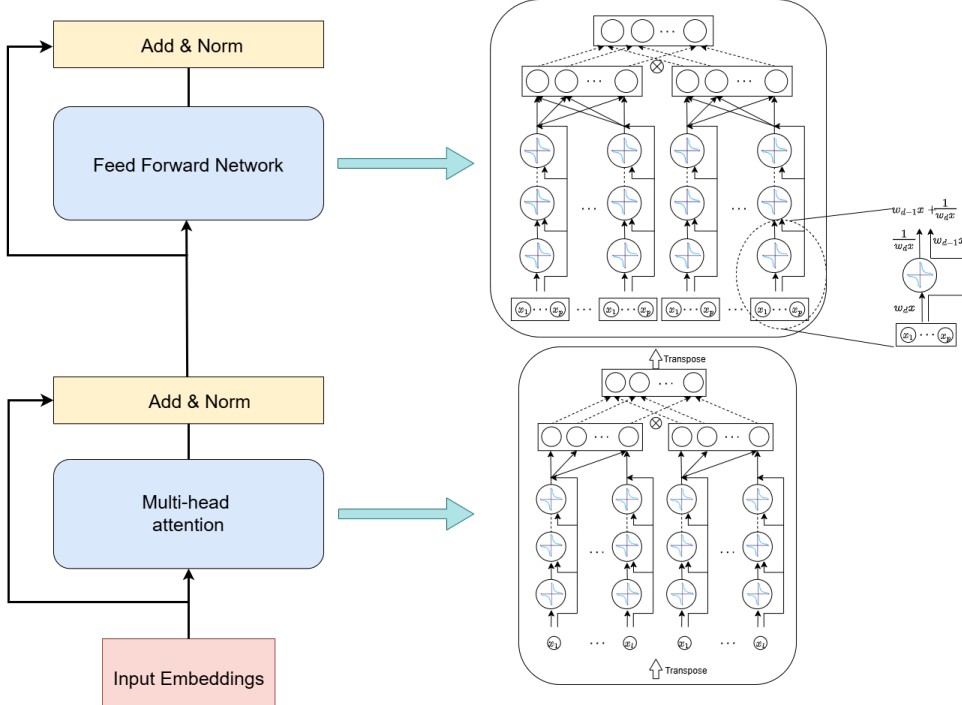

*Figure 1.* Above we see a Transformer block consisting of attention and FFN layers. We propose candidate CoFrNet architectures for Transformer (causal) attention and FFN layers. The circles with the blue curves denote the $\frac{1}{x}$ non-linearity in our architectures. The zoomed out image on the far right shows the mapping between the pictorial representation and the actual equations. Details of the architectures are discussed in section 4.

## 3. Related Work

A brief historical perspective on artificial neural networks is provided in the appendix. Turning our focus to language modeling with neural networks, Recurrent Neural Networks (RNNs), a class of networks with recurrent connections where the output of a neuron at a time step is fed to the input of the neuron at the next time step, were successful in many tasks such as machine translation (Sutskever et al., 2014) and language modeling (Jozefowicz et al., 2016). The encoder-decoder Transformer model proposed in (Vaswani et al., 2017), avoids recurrence and relies on attention alone to draw dependencies between the input and output, and these models have revolutionized language modeling. The two early successful transformer architectures that have led to a series of models include the Generative Pre-trained Transformer (GPT) (Radford et al., 2018) and Bidirectional Encoder Representations from Transformers (BERT) (Devlin et al., 2019). These pre-trained models can be then *fine-tuned* on relatively small datasets (Raffel et al., 2020; Chung et al., 2024; Wang et al., 2022) leading to good performance on even unseen tasks. Transformer models, because of their uncompressed view on the entire sequence, show measurable improvement in performance over RNNs, but the attention mechanism scales quadratically with sequence length, as opposed to the linear time generation complexity

of RNNs. Given this multiple approximations have been proposed to model attention in Transformers more efficiently. Works such as Synthesizer (Tay et al., 2021) and Linformer (Wang et al., 2020) try to make attention linear complexity, while Mixture-of-depths attention (Gadhikar et al., 2024) and Sliding Window attention (Fu et al., 2025) limit the number of attended tokens in a sequence. Slim attention (Graef & Wasielewski, 2025) does away with the value parameter matrix and models it as a function of the key matrix. Multi-query attention (Shazeer, 2019) and its generalization Grouped Query attention (Joshua et al., 2023) limit the number of distinct keys thus reducing parameter count and increasing efficiency. Sparse attention approaches (Zaheer et al., 2024) typically attend to local context and sparsely to further away tokens (a.k.a. global context).

Aside from RNNs and Transformers, State-Space Models (SSMs) have also been quite popular. Models such as S4 (Gu et al., 2022) and Mamba (Gu & Dao, 2024) are recurrent like RNNs, but can handle long range dependencies. The latter selectively propagates information based on the current token making it closer to the modeling power of Transformers, while scaling linearly in sequence length. More recently, Diffusion Models inspired by non-equilibrium statistical physics (Sohl-Dickstein et al., 2015) have gained traction. The attractive aspect of these models is that gen-

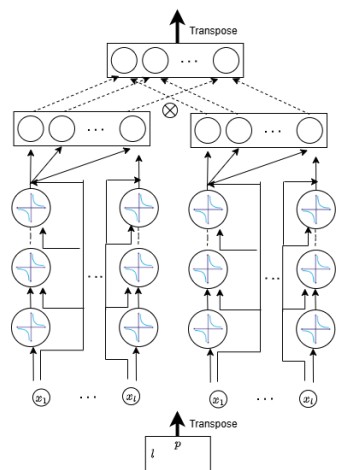 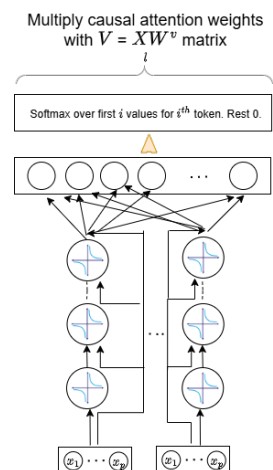

*Figure 2.* Two CoFrNet architectures to simulate attention a.k.a. causal token-token mixing. For the left architecture (CAttnU) a transpose is taken of the dimension × sequence length part of the input tensor and the output is transposed back to make it consistent with the later layers. The transpose makes the tokens mix, while upper triangular connections in the second to last layer in the architecture as well as the restricted structure of the ladders make sure information is *only* shared from previous tokens to following tokens and not bi-directionally (a.k.a. causal sharing). It consists of two ensembles of univariate CoFrNet ladders each of which then have an upper triangular linear layer on top. The representations formed are then element wise multiplied to form the final representation. The element wise multiplication produces interaction terms that otherwise would not occur, significantly enhancing representation power without compromising the causal information flow. The right architecture (CAttnM) we do not transpose the input. We use $L$ CoFrNet ladders that get mapped to a sequence length size embedding which corresponds to attention weights for that token. To maintain causality attention weights are computed only over the prior tokens. These then like in standard attention are used to weight the embeddings in the (value) $V$ matrix.

eration does not have to be auto-regressive and can happen in parallel. In (Sahoo et al., 2024a), the authors propose a simple Masked Diffusion Language Model (MDLM) using an effective training recipe that narrows the gap of diffusion and autoregressive methods in language modeling. Nonetheless, Transformers are still the state-of-the-art in language generation and hence we chose to modify critical components of this architecture.

## 4. Methodology

### 4.1. Architectures

We now describe our novel continued fraction architectures that can potentially be used instead of attention and FFN layers in Transformer blocks.

*Table 1.* Scale of parameters for different architectural components. Here $\alpha >> 1$ is expansion factor for FFNs in Transformer blocks. The savings in parameters when replacing FFNs can be significantly high as low $d$ and $L$ values are typically sufficient for competitive performance. For attention replacement the savings can be high if $l$ is similar order of magnitude to $p$, which is seen in many architectures (viz. GPT, Llama, etc.).

| Attention | CAttnU | CAttnM | FFN | Cffn |
|---|---|---|---|---|
| $4p^2$ | $l(2d + l + 1)$ | $L(p + l) + p^2$ | $2\alpha p^2$ | $2Lp(d + 1)$ |

#### 4.1.1. REPLACEMENT FOR ATTENTION

In Figure 2, we see two potential architectures that perform causal token-token mixing. In the *left architecture*, we take

a transpose of the input tensor relative to the embedding dimension and sequence length, which has been done in MLP-Mixer type models (Tolstikhin et al., 2021) employed for supervised problems. However, mixing a dimension across tokens arbitrarily will lead to *non-causal* training as the model will get trained assuming access to tokens that follow a given token. To handle this we have univariate ladders – note an input now is a particular dimension across all $l$ tokens – where, $x_1$ will get different dimensions of the first token in the sequence, $x_2$ will get different dimensions of the second token in the sequence and so on. Hence, $x_1$ can affect all tokens, but $x_2$ can affect all but $x_1$. This is why we have upper triangular linear layer in each ensemble of the architecture. Note that having $p$-variate ladders would break the causal transfer even with upper triangular linear layers as output from each of the ladders would be a function of all tokens. Hence, we have this restricted structure to maintain the causal information constraints else generations are incoherent. We then do element wise multiplication to obtain cross-terms in the variables as the ladders are univariate leading to richer representations. In particular, if depth of the ensembles $d = 2$, where $w_0^{(1)}$, $w_0^{(2)}$ are parameter vectors at depth 1 and $w_1^{(1)}$, $w_1^{(2)}$ are parameter vectors at depth 2 for the left and right ensembles respectively, then if $\odot$ implies element-wise multiplication and $\circ - 1$ implies element-wise reciprocal we would get:

$$y_1 = w_0^{(1)} \odot x + \left(w_1^{(1)} \odot x\right)^{\circ-1} \quad \text{and} \quad y_2 = w_0^{(2)} \odot x +$$

$(w_1^{(2)} \odot x)^{\circ -1}$.

Let $U_1$ and $U_2$ denote upper triangular parameter matrices then, $O = U_1 y_1 \odot U_2 y_2$. $O$ is the $l$ dimensional output produced per input $x$. In our case we will get $p$ such outputs. The tensor containing these $p$ outputs is then transposed back to get a $l \times p$ tensor, which later layers expect.

Now considering the *right architecture* with two ladders (i.e. $L = 2$) of depth 2, a $L \times l$ (full) parameter matrix $F$ and Csoftmax to denote softmax applied causally (i.e. $i^{\text{th}}$ token is a convex combination of the first $i - 1$ tokens) with notation from above we have attention weights given by,

$A = \text{Csoftmax}([y_1, y_2]F)$, where in this case $y_1 = w_0^{(1)^T} x + \left(w_1^{(1)^T} x\right)^{-1}$ and $y_2 = w_0^{(2)^T} x + \left(w_1^{(2)^T} x\right)^{-1}$ as no transpose of the input tensor is taken and hence $x$, $w$ are $p$ dimensional. If $V = XW^v$ denotes a value matrix like in standard attention where $W^v$ is a $p \times p$ parameter matrix, then the output $O$ is given by: $O = AV$, which would be $l \times p$ tensor.

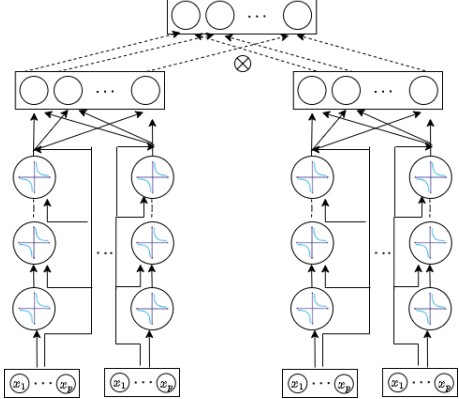

*Figure 3.* CoFrNet architecture to simulate FFNs – Cffn – in a transformer block. Here again two ensembles are used each consisting of specified number of $p$-variate ladders. Here no transpose is taken and hence feature mixing in either direction does not interfere with causal generation which is why we have a linear layer above each ensemble. We also element wise multiply the representations coming out of the linear layer of each ensemble for higher expressivity. Again the collapsed implementation is described in section 4.2.

### 4.1.2. Replacement for FFNs

For FFNs we simply require feature mixing so no transpose is taken and all features can mix. Hence, we create ensembles of $p$-variate ladders with a linear layer at the end as seen in Figure 3.

Note that here one could have an arbitrary number of ladders in each ensemble and one projects to $p$ dimensions using the linear layer. We again multiply the representations coming out of the linear layers for richer representation learning.

Expressions depicting the scale of parameters of different architectural components are shown in Table 1. As can be seen the *number of parameters are linear in $p$ as opposed to quadratic*.

### 4.2. Architecture for Continued Fraction Ensembles and Continuant-Based Implementation

The common element in the architectures in Figures 2 and 3 is a linear combination of an ensemble of CoFrNet ladders. This subsection describes how we implement these linear combinations of ladders using the continuants introduced in Section 2.

**Architecture** Let us denote by $y \in \mathbb{R}^q$ the output of a linear combination of $L$ ladders, where in general $q$ could be different from the input dimension $p$. We use a superscript $j$ to denote the partial denominators $a_0^{(j)}, \ldots, a_d^{(j)}$ corresponding to the $j$th ladder, where $a_k^{(j)} = w_k^{(j)} x$. Then based on (2), the $i$th output component $y_i$ is given by

$$y_i = \sum_{j=1}^{L} v_{ij} \left( a_0^{(j)} + \tilde{f}(a^{(j)}) \right)$$

$$= \sum_{j=1}^{L} v_{ij} w_0^{(j)} x + \sum_{j=1}^{L} v_{ij} \tilde{f}(a^{(j)}), \tag{7}$$

where $v_{ij}$ are the coefficients of the linear combination. Since the composition of two linear functions is also linear, we may simplify the first term on the right-hand side of (7) to yield

$$y_i = u_i x + \sum_{j=1}^{L} v_{ij} \tilde{f}(a^{(j)}),$$

where $u_i = \sum_{j=1}^{L} v_{ij} w_0^{(j)}$ is the parameter vector of the overall linear function. Let $U$ be the matrix with rows $u_i$, $i = 1, \ldots, q$, $V$ the matrix with entries $v_{ij}$, and $W^{(j)}$ the matrix with rows $w_k^{(j)}$, $j = 1, \ldots, d$. We may then express the overall computation from $x$ to $y$ as

$$y = Ux + Vz, \ z_j = \tilde{f}(a^{(j)}), \ a^{(j)} = W^{(j)} x, \ j = 1, \ldots, L. \tag{8}$$

Based on (8), we implement a linear combination of ladders using the architecture shown in Figure 4. At the far left is a linear layer parameterized by $U$ that directly connects input $x$ to output $y$. To the right are $L$ ladders, where for each ladder $j$, a linear layer parameterized by $W^{(j)}$ first computes the partial denominators $a^{(j)}$ before the continued fraction is computed by the "CF" layer. The continued fraction outputs $z_j$ are fed to a linear layer parameterized by $V$, whose output is added to yield $y$.

**Continuant implementation** We use the continuants representation from Section 2 to compute continued fractions in the CF layer. Specifically, continuants $K_0, K_1, \ldots, K_d$ are

first computed using the recursion in (4), (5). The continued fraction output $\tilde{f}(a^{(j)})$ is then given by the ratio of $K_{d-1}$ and $K_d$ in (6). The following result shows that the *gradient* of $\tilde{f}(a^{(j)})$ is also given by ratios of continuants.

**Proposition 4.1.** *The partial derivatives of continued fraction $\tilde{f}(a)$ defined in (2) are given by*

$$\frac{\partial \tilde{f}(a)}{\partial a_k} = (-1)^k \left( \frac{K_{d-k}(a_{k+1}, \ldots, a_d)}{K_d(a_1, \ldots, a_d)} \right)^2, \ k = 1, \ldots, d. \tag{9}$$

*Proof.* Using equations (2) and (3) we get,

$$\frac{\partial \tilde{f}(a)}{\partial a_k} = \frac{\partial}{\partial a_k} \left( f(a_0, a) - a_0 \right) = \frac{\partial}{\partial a_k} \frac{K_{d+1}(a_0, \ldots, a_d)}{K_d(a_1, \ldots, a_d)} - 0$$

for $k = 1, \ldots, d$. We then invoke Lemma 2 stated in the appendix. □

To take advantage of Proposition 4.1, we implement the CF layer in Figure 4 as a custom PyTorch function (`torch.autograd.Function`). This allows the continuants $K_0, \ldots, K_d$, as well as the reciprocal $1/K_d$, to be computed once during the forward pass and saved for the backward pass. Then to compute the gradient, it suffices to multiply $1/K_d$ by other continuants, square the ratios, and change some signs.

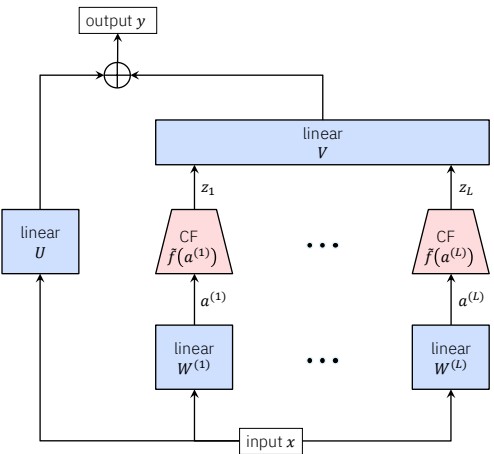

*Figure 4.* Architecture for implementing a linear combination of CoFrNet ladders (CF stands for continued fraction).

**Advantages** Using continuants to compute each continued fraction $\tilde{f}(a^{(j)})$ (6) and its gradient (9) requires only one division, by the same quantity $K_d$. As noted above, the reciprocal $1/K_d$ can be computed once and then reused in all ratios of continuants that are required. As seen from (5), all continuants up to $K_d$ can be computed recursively through $O(d)$ multiplications and additions. This continuants approach yields a major improvement in efficiency over the "literal" approach taken in the original CoFrNet

work (Puri et al., 2021), which performs one division per layer following the standard representation of a continued fraction (1). The reduction from $d$ divisions to 1 is especially significant when ladders are made deep. It applies to both inference and training, since backpropagation through a standard PyTorch implementation of (1) also requires $d$ divisions. It is widely known that *divisions are significantly more expensive in current hardware* — typically an order of magnitude slower — than multiplications or additions. Moreover, having to divide just once can result in *better numerical stability*.

**Avoiding poles and clipping** Equation 6 shows that a continued fraction is equivalent to a rational function, and hence it can suffer from divergence when the denominator $K_d$ goes to zero (these locations are known as *poles* in the context of rational functions). We mitigate this issue using a similar approach as (Puri et al., 2021), namely changing the denominator from $K_d$ to $\text{sgn}(K_d) \max(|K_d|, \epsilon)$ to ensure that it has absolute value at least $\epsilon > 0$. Importantly however, this modification is done only once to $K_d$ as opposed to before every one of the $d$ divisions in (Puri et al., 2021). This may result in less loss of representation power compared to (Puri et al., 2021).

We also maintain the minimum and maximum values that each ladder produces during training. During testing we project or clip predictions to lie in this range so that outputs far away from those seen during training are not produced thus guarding against outlier test predictions.

## 5. Experiments

### 5.1. Setup

We now perform experiments, where we compare with GPT2-xl (G-xl) 1.5B first pre-trained on OpenWebText (OWT) (Gokaslan et al., 2019) and then on the Gneiss-Web 35B (GW) (Gohari et al., 2025) datasets. We compare with three variants of ours i) CoFrGeNet-F (C-F), where the FFN is replaced by CoFrNet, ii) CoFrGeNet-A (C-A), where the attention is replaced by CoFrNet and iii) CoFr-GeNet (C), where both FFN and attention are replaced. We report results with the CAttnM architecture when attention is replaced as it led to slightly better results than CAttnU in many cases. We also compare with Dense Synthesizer (S-D) (Tay et al., 2021) which is closest to our CAttnM architecture and an established sparse attention approach (S-A) (Zaheer et al., 2024). To test the efficacy of CoFrNet on a different architecture we experiment with Llama-3.2B pre-trained on the docling data mix (Team, 2024) of 2T tokens. The data mix contains web (DCLM2, DCLM3Plus (Li et al., 2024)), multilingual (FineWeb-2-edu (Lozhkov et al., 2024)), code (Starcoder, stack-edu (Allal et al., 2025)), math (Finemath (Allal et al., 2025), Infiwebmath (Han et al.,

2024), opc-fineweb-math-corpus (Huang et al., 2024)) and synthetic data (Cosmopedia (Ben Allal et al., 2024)), which is heavily used to train models for diverse document understanding. The Llama models already use an efficient form of attention namely Grouped Query Attention (GQA) and hence are a natural efficient attention baseline when replacing components of Llama.

**Evaluations:** We report perplexity on Penn Tree Bank (PTB) (Marcus et al., 1993), Wikitext2 (Merity et al., 2017), Wikitext103 (Merity et al., 2017), Lambada (Paperno et al., 2016), AgNews (Zhang et al., 2015) and One Billion Words (LM1B) (Chelba et al., 2014) datasets. We use a stride of 512 for wikitext2, wikitext103 as recommended in these works. For all the other datasets, we use a stride of 256. We then fine tune our models on GLUE (Wang et al., 2019) (classification) tasks and compare accuracies as done in previous works (Sahoo et al., 2024b). We average results over five runs. We also compare parameter counts, train time

*Table 2.* Downstream task accuracies (best results bolded) on GLUE benchmark after finetuning. The first column is the pre-training dataset. Stds are reported in Table 9 in the appendix.

| Model | MNLI | QQP | QNLI | SST2 | COLA | MRPC | RTE | WNLI |
|---|---|---|---|---|---|---|---|---|
| OWT | | | | | | | | |
| G-xl (1.5B) | 86.89 | 88.93 | 91.35 | 93.56 | 81.78 | 79.83 | 60.27 | 58.28 |
| C-F (972M) | **87.24** | **89.95** | **91.87** | **94.12** | **82.57** | **80.17** | **61.36** | **58.30** |
| C-A (1.21B) | 86.94 | 89.31 | 91.74 | 93.83 | 81.77 | 79.89 | 60.91 | 58.28 |
| C (790M) | 87.11 | 89.36 | 91.79 | 93.91 | 81.97 | 79.93 | 61.25 | 58.29 |
| S-D (1.2B) | 84.93 | 86.82 | 90.13 | 91.34 | 80.15 | 77.95 | 59.83 | 58.28 |
| S-A (1.21B) | 85.27 | 86.38 | 90.93 | 92.72 | 80.76 | 77.42 | 59.36 | 58.27 |
| GW | | | | | | | | |
| G-xl (1.5B) | 78.28 | 86.83 | **82.93** | 91.82 | 74.18 | 77.72 | 60.19 | **58.33** |
| C-F (972M) | **79.23** | **87.24** | 82.69 | **92.38** | **74.79** | **78.04** | **61.37** | **58.33** |
| C-A (1.21B) | 78.42 | 86.17 | 82.51 | 91.86 | 74.15 | 77.37 | 60.85 | **58.33** |
| C (790M) | 79.05 | 86.98 | 82.12 | 92.13 | 74.38 | 77.95 | 61.11 | **58.33** |
| S-D (1.2B) | 77.56 | 86.35 | 80.38 | 91.25 | 73.27 | 76.73 | 59.26 | 58.24 |
| S-A (1.21B) | 77.67 | 86.41 | 80.77 | 91.16 | 72.83 | 76.62 | 59.39 | 58.28 |

and (per-sample) inference time. We show how the continuants version leads to better train and inference time when compared with the standard implementation of CoFrNets with the improvement mainly attributable to the reduced number of divisions. We provide randomly chosen generations for our variants and GPT2-xl in the appendix. For Llama-3.2B, we evaluate on openbookqa (Mihaylov et al., 2018), piqa (Bisk et al., 2020), arc-easy (Clark et al., 2018), winogrande (win, 2019), hellaswag (Zellers et al., 2019), lambada open AI (Radford et al., 2018), boolq (Christopher et al., 2019) and sciq (Welbl et al., 2017) which cover open domain Q&A, reasoning and text understanding tasks. We also report throughput and training time.

We also plot attention maps to qualitatively evaluate our

attention replacements. Additionally, we assess the fluency, coherence and repetition rate of our generations.

*Table 3.* Perplexities of the different variants with GPT2-xl.

| Model | PTB | Wikitxt2 | Lbda | AgNews | Lm1b | Wiki103 |
|---|---|---|---|---|---|---|
| OWT | | | | | | |
| G-xl (1.5B) | 30.12 | 18.30 | 8.66 | 37.13 | 41.20 | 17.50 |
| C-F (972M) | **29.94** | **17.09** | **8.15** | **35.72** | **40.11** | **16.17** |
| C-A (1.21B) | 30.02 | 18.22 | 8.54 | 37.02 | 41.03 | 17.26 |
| C (790M) | 30.03 | 17.96 | 8.55 | 36.47 | 40.86 | 17.17 |
| S-D (1.2B) | 31.47 | 19.35 | 9.92 | 39.84 | 41.94 | 18.91 |
| S-A (1.21B) | 31.23 | 18.78 | 9.13 | 38.82 | 42.05 | 18.82 |
| GW | | | | | | |
| G-xl (1.5B) | 29.07 | 19.12 | 31.78 | 45.62 | 52.36 | 18.93 |
| C-F (972M) | 29.83 | **18.08** | **30.55** | **41.77** | **46.59** | **18.11** |
| C-A (1.21B) | **28.89** | 18.77 | 30.98 | 43.91 | 48.37 | 18.67 |
| C (790M) | 29.08 | 18.29 | 30.71 | 42.55 | 48.01 | 18.42 |
| S-D (1.2B) | 30.83 | 19.25 | 31.92 | 46.81 | 52.99 | 19.03 |
| S-A (1.21B) | 29.36 | 18.95 | 31.23 | 46.38 | 52.83 | 19.45 |

**Parameter Settings:** For pre-training GPT2-xl we use the recommended settings in `https://github.com/karpathy/nanoGPT` where, the learning rate is $6 \times 10^{-4}$, weight decay is $0.1$, no dropout and maximum iterations is $600K$. For sparse attention (Sparse Attn) we set $g = 1$, $w = 3$ and $r$ is set to roughly match the number parameters in our CoFrGeNet-A variant for a fair comparison. The values of $g$ and $w$ were set based on experiments conducted in (Zaheer et al., 2024) as those produced the best results. For both Synthesizer-D and Sparse Attn we apply a lower triangular mask to the attention weights matrix so as to make the models amenable for auto-regressive generation.

For fine tuning the GPT2-xl model learning rate is $0.25 \times 10^{-4}$, batch size is $64$ and no dropout. This is the same for the baselines. For our models the learning rate was $0.125 \times 10^{-4}$ with other parameters being the same. These learning rates produced the best results for the respective models.

The Llama variants we pre-train for about 2M iterations. The initial learning rate is $3 \times 10^{-4}$ and follows an annealing schedule with no dropout. Adam optimizer is used for both model variants.

For CoFrNets we set $\epsilon = 0.01$. For Cffn architecture we have two ensembles where we make sure that they have even-odd depths. That is if the first ensemble has depth $d$, then the next one has depth $d + 1$. We find this gives better results which, may be attributable to the fact that even and odd convergents typically converge to different parts of the space. This may cover the function space better. Given this we experiment with $d$ equal to $1, 3, 5, 7$ and widths (i.e. number of ladders in each ensemble) also taking the same values when replacing FFNs. We try the same depths and widths when replacing attention.

**Training Schedule:** We employ a dyadic parameter update schedule for our CoFrGeNet components. More specifically, we update only the linear component starting from iteration one, where parameters at higher depths are frozen. Then after half the iterations are done we start updating also

*Table 4.* Training time and inference time. $C_B$ is our basic implementation not using continuants. As can be seen using the continuants formalism speeds up training and inference.

| Data | Model | Train Time (hrs) | Inf. Time ($\mu$s) |
|---|---|---|---|
| OWT | G-xl | 190 | 643.93±1.73 |
| | C-F | 186 | 627.48±1.85 |
| | C-A | 186 | 638.26±1.76 |
| | C | 178 | 628.73±1.66 |
| | $C_B$ | 203 | 5898.72±3.91 |
| GW | G-xl | 413 | 638.26±2.73 |
| | C-F | 397 | 627.34±1.65 |
| | C-A | 396 | 625.86±1.78 |
| | C | 387 | 619.78±1.49 |
| | $C_B$ | 424 | 5877.87±4.52 |

the first layer parameters. Then after $\frac{3}{4}^{\text{th}}$ the number of iterations we start updating the depth two parameters and so on. Essentially, depth $i$ parameters are updated for $\frac{t}{2^i}$ number of iterations where $t$ is the total number of iterations. We find that this leads to stable training of our architectures as opposed to training all parameters from the start.

**Hardware:** We pre-trained the GPT models using 16 H100 GPUs and distributed data parallel (ddp) training. Fine tuning was done using a single A100 GPU for each model. Also inference times were computed for all models using a single A100 GPU. The Llama models were pre-trained using 128 H100 GPUs with fully sharded distributed data parallel (fsdp) training.

*Table 5.* Perplexities of CoFrGeNet (GPT2-xl) variants with (top number) and without (below number) incremental training. As seen our training schedule has significant impact.

| Model | PTB | Wikitxt2 | Lbda | AgNews | Lm1b | Wiki103 |
|---|---|---|---|---|---|---|
| OWT | | | | | | |
| C-F (972M) | **29.94** | **17.09** | **8.15** | **35.72** | **40.11** | **16.17** |
| | 33.72 | 26.71 | 12.56 | 42.18 | 47.28 | 22.65 |
| C-A (1.21B) | 30.02 | 18.22 | 8.54 | 37.02 | 41.03 | 17.26 |
| | 38.24 | 21.82 | 10.92 | 45.52 | 46.21 | 24.25 |
| C (790M) | 30.03 | 17.96 | 8.55 | 36.47 | 40.86 | 17.17 |
| | 36.77 | 23.87 | 15.23 | 42.72 | 49.44 | 23.33 |
| GW | | | | | | |
| C-F (972M) | 29.83 | **18.08** | **30.55** | **41.77** | **46.59** | **18.11** |
| | 35.88 | 25.55 | 37.33 | 45.46 | 49.53 | 20.44 |
| C-A (1.21B) | **28.89** | 18.77 | 30.98 | 43.91 | 48.37 | 18.67 |
| | 33.71 | 23.72 | 36.28 | 45.29 | 52.51 | 21.67 |
| C (790M) | 29.08 | 18.29 | 30.71 | 42.55 | 48.01 | 18.42 |
| | 34.22 | 22.98 | 36.23 | 44.39 | 51.91 | 21.67 |

*Table 6.* Zero-shot accuracies on open domain Q&A, reasoning and text understanding tasks. The docling data mix of 2 trillion tokens was used for pre-training.

| Model | Opqa | Piqa | Arc | Wino | Hswag | Lbda | Boolq | Sciq |
|---|---|---|---|---|---|---|---|---|
| Llama (3.2B) | .282 | .76 | .77 | **.654** | **.503** | **.581** | **.691** | **.941** |
| C-F (1.9B) | .292 | **.764** | .765 | .643 | .482 | **.581** | .659 | **.941** |
| C-A (2.5B) | .304 | .752 | .757 | .646 | .463 | .575 | .633 | .914 |
| C (1.7B) | .283 | .751 | .751 | .64 | .464 | .571 | .633 | .907 |
| Mamba-2 (3.2B) | **.324** | .761 | .768 | .615 | .486 | .548 | .655 | .919 |

## 5.2. Results

One of the main ways of evaluating if a generative model has learnt good representations is to test it on downstream tasks. In Table 2 we evaluate how our models perform w.r.t. GPT2-xl on GLUE tasks. We observe that our models are much smaller – sizes are mentioned next to the names in column two – yet are better in performance in most cases

to the original GPT2-xl model. In fact, they are also better than the linear attention and sparse attention baselines being similar or smaller size. For the Sparse Attn baseline the size reflects the sparsity level or the number of non-zeros. CoFrGeNet-F seems to have the best performance amongst all the variants in most cases. In Table 3, we evaluate how confident the model is in its generations. We see in Table 3 that again our models are better than GPT2-xl and the efficient attention baselines. Here again CoFrGeNet-F seems to have the best perplexity in most cases consistent with the fine tuning performance. In Table 4, we compare training

*Table 7.* Throughput for Llama-3.2B and our variants.

| Model | Tokens/day | Train Time (days) |
|---|---|---|
| Llama (3.2B) | 235B | 8.5 |
| C-F (1.9B) | 288B | 7 |
| C-A (2.5B) | 250B | 8 |
| C (1.7B) | 315B | 6.4 |

and inference times of our models and GPT2-xl. Here we add an additional model CoFrGeNet$_B$ which is the same architecture as CoFrGeNet, but implemented as multi-layer ladders as done in (Puri et al., 2021), without exploiting the continuants formalism. This means a division operation has to be done at every layer of the ladder while training and inferring. As can be seen the training for the continuants version is faster, with inference being almost an order of magnitude faster. In Table 5, we compare the perplexities of our trained models with and without our custom training schedule. As can be seen our training schedule leads to much better performing models as it stabilizes training.

*Table 8.* Qualitative metrics to evaluate generation quality of GPT2-xl and CoFrGeNet variants using Claude Opus 4.6 as a judge. Results were averaged over 100 (random) generations for models pre-trained on OWT. Claude was instructed to produce scores between 0 and 1 for fluency and coherence for the generated texts where 1 implied perfection. Claude was instructed to measure the number of times each phrase is repeated in the text and report the average number of repetitions. The repetition rate relative (RRR) to GPT2-xl for the different models is also reported below. As can be seen the generation quality for all the models is comparable (CoFrGeNet-F is best), however our models are smaller.

| Model | Fluency↑ | Coherence↑ | RRR↓ |
|---|---|---|---|
| G-xl (1.5B) | 0.83 | 0.75 | 1 |
| C-F (972M) | 0.85 | 0.8 | 0.97 |
| C-A (1.21B) | 0.84 | 0.78 | 1.06 |
| C (790M) | 0.83 | 0.77 | 1.08 |

In Table 6, we observe similar qualitative behavior for the Llama models even when tested on diverse tasks ranging from open domain Q&A to reasoning, where CoFrGeNet-F is the best on majority of these tasks, while the other variants are still competitive with the original Llama model. The throughputs are observed in Table 7. We see that our variants are faster than the original Llama where, CoFrGeNet-F and CoFrGeNet take as much as a couple of days less.

In Figure 5, we see that our architectures for replacing attention focus on meaningful tokens. While in Table 8, we

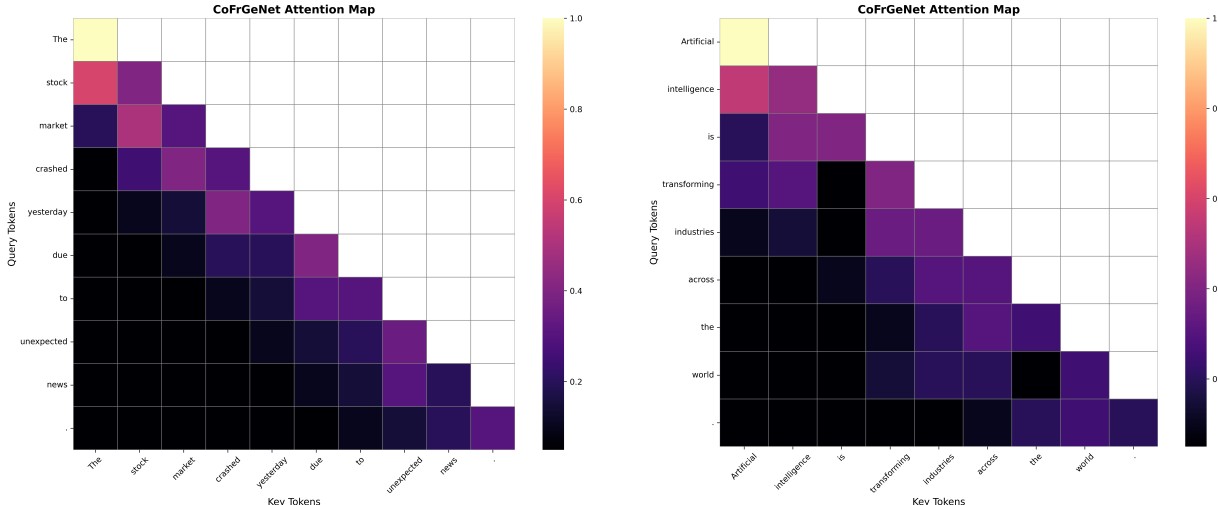

*Figure 5.* Last layer attention map for CoFrGeNet for two randomly picked sentences from OWT when replacing attention and FFNs in GPT2-xl. The attention maps capture meaningful relationships such as "crashed" depends on "market" in the left figure and "transforming" depends on "artificial" and "intelligence".

see that our generations are of comparable quality to the original Transformer model.

These results suggest that across model architectures and tasks our architectural modifications lead to competitive models that are parameter efficient.

## 6. Discussion

We have proposed novel continued fraction inspired architectures as replacements for attention and FFNs in transformer blocks. This new interesting function class can learn accurate, compact models that are also efficient to train and infer. Our continuant based gradient derivation and implementation facilitated these benefits over and above optimizing these architectures by backpropagating through the layers using standard Pytorch functionalities as done previously (Puri et al., 2021). The custom training schedule for CoFrGeNet specific parameters further helped stabilize and improve performance. In the future, it would be interesting to experiment with other open architectures such as Mamba as well as Mixture-Of-Experts kind of architectures. Inventing new and better CoFrNet architectures for attention and FFNs beyond those proposed in this work is another interesting direction. Also building custom Triton Kernels (Tillet et al., 2019) for our components to further speedup training and inference might be a worthwhile future effort.

As such we believe we have laid the groundwork for continued fraction inspired generative architectures. This could lead to small, efficient to train and accurate generative models across applications and industries. In a way this could further democratize AI as entities with fewer resources could also pre-train good quality models.

Our use of divisions as the non-linearity distinguishes our architecture from others. However, divisions are more expensive than matrix multiplications and additions in modern digital hardware and this is a drawback. We are working towards hardware as well as software solutions for this. For instance, outsourcing divisions to FPGA hardware is something we are seeing initial promise in. Also writing divisions in some other form may be interesting to explore from a software side. Also, there are no implicit safety guards for these models similar to other architectures and so they are susceptible to hallucinations, adversarial attacks and the likes. We hope future research exploiting the specific functional form can implicitly address some of these challenges, which we believe could be very exciting.

## Impact Statement

We believe our work advances the important topic of building language models and small language models (SLMs) in particular. SLMs are having huge impact on industry and society at large. Our work proposes a novel way to build these whose implications can be further studied as this work is disseminated to society. As mentioned in the discussion there seem to be no implicit safety guards for these models similar to other architectures and so they are susceptible to hallucinations, adversarial attacks and the likes. However, given their compact and inherently interpretable nature at the component level it may be possible to design more robust and safer systems in the future.

## Acknowledgements

We would like to thank David Cox for pointing us towards the docling data mix as well as the GW dataset and making sure we have resources to run the experiments. We would also like to thank Hajar Gohari's team and Ahmed Nassar for details on these datasets as well as corresponding models and training frameworks. Also special thanks to Kush Varshney, Sriram Raghavan and Ruchir Puri for supporting this effort.

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

## A. Brief Historical Perspective

One of the starting points of artificial neural networks was in the mathematical model of biological neurons known as *artificial neurons* or McColluch-Pitts Neurons proposed in (McCulloch & Pitts, 1943). These artificial neurons were remarkably similar to the elements used in modern neural networks, in that their output is a thresholded weighted sum of their inputs. The Multi Layer Perceptron (MLP) (Rosenblatt, 1958) used multiple layers of neurons with input, hidden and output layers as a simplified model of the nervous system. The Group Method of Data Handling (GMDH) (Ivakhnenko, 1971) trained a network with an MLP-type structure but each neuron in the network implements a polynomial function of a few input variable, and this was used to train a network that is 8 layers deep.

However, practical learning of networks was made easier after error backpropagation was published (Linnainmaa, 1976) and demonstrated for weight update and learning representation in neural networks (Rumelhart et al., 1986).

## B. Lemma 2 (Puri et al., 2021)

We have

$$\frac{\partial}{\partial a_k} \frac{K_{d+1}(a_0, \ldots, a_d)}{K_d(a_1, \ldots, a_d)} = (-1)^k \left( \frac{K_{d-k}(a_{k+1}, \ldots, a_d)}{K_d(a_1, \ldots, a_d)} \right)^2.$$

*Proof.* To compute the partial derivative of the ratio of continuants above, we first determine the partial derivative of a single continuant $K_k(a_1, \ldots, a_k)$ with respect to $a_l$, $l = 1, \ldots, k$. We use the representation of $K_k$ as the determinant of the following tridiagonal matrix:

$$K_k(a_1, \ldots, a_k) = \det \begin{bmatrix} a_1 & 1 & & \\ -1 & a_2 & \ddots & \\ & \ddots & \ddots & 1 \\ & & -1 & a_k \end{bmatrix}. \tag{10}$$

The partial derivatives of a determinant with respect to the matrix entries are given by the *cofactor* matrix:

$$\frac{\partial \det A}{\partial A_{ij}} = \mathrm{co}(A)_{ij},$$

where $\mathrm{co}(A)_{ij} = (-1)^{i+j} M_{ij}$ and $M_{ij}$ is the $(i, j)$-minor of $A$. In the present case, with $A$ as the matrix in (10), we require partial derivatives with respect to the diagonal entries. Hence

$$\frac{\partial K_k(a_1, \ldots, a_k)}{\partial a_l} = M_{ll}.$$

In deleting the $l$th row and column from $A$ to compute $M_{ll}$, we obtain a block-diagonal matrix where the two blocks are tridiagonal and correspond to $a_1, \ldots, a_{l-1}$ and $a_{l+1}, \ldots, a_k$. Applying (10) to these blocks thus yields

$$\frac{\partial K_k(a_1, \ldots, a_k)}{\partial a_l} = K_{l-1}(a_1, \ldots, a_{l-1}) K_{k-l}(a_{l+1}, \ldots, a_k). \tag{11}$$

Returning to the ratio of continuants in the lemma, we use the quotient rule for differentiation and (11) to obtain

$$\begin{aligned} \frac{\partial}{\partial a_k} \frac{K_{d+1}(a_0, \ldots, a_d)}{K_d(a_1, \ldots, a_d)} &= \frac{1}{K_d(a_1, \ldots, a_d)^2} \left( \frac{\partial K_{d+1}(a_0, \ldots, a_d)}{\partial a_k} K_d(a_1, \ldots, a_d) \right. \\ &\quad \left. - K_{d+1}(a_0, \ldots, a_d) \frac{\partial K_d(a_1, \ldots, a_d)}{\partial a_k} \right) \\ &= \frac{K_{d-k}(a_{k+1}, \ldots, a_d)}{K_d(a_1, \ldots, a_d)^2} \left( K_k(a_0, \ldots, a_{k-1}) K_d(a_1, \ldots, a_d) \right. \\ &\quad \left. - K_{d+1}(a_0, \ldots, a_d) K_{k-1}(a_1, \ldots, a_{k-1}) \right). \end{aligned} \tag{12}$$

We focus on the quantity

$$K_k(a_0, \ldots, a_{k-1}) K_d(a_1, \ldots, a_d) - K_{k-1}(a_1, \ldots, a_{k-1}) K_{d+1}(a_0, \ldots, a_d) \tag{13}$$

in (12). For $k = 0$ (and taking $K_{-1} = 0$), this reduces to $K_d(a_1, \ldots, a_d)$. Equation (12) then gives

$$\frac{\partial}{\partial a_0} \frac{K_{d+1}(a_0, \ldots, a_d)}{K_d(a_1, \ldots, a_d)} = \left( \frac{K_d(a_1, \ldots, a_d)}{K_d(a_1, \ldots, a_d)} \right)^2 = 1,$$

in agreement with the fact that $a_0$ appears only as the leading term in (3). For $k = 1$, (13) becomes

$$a_0 K_d(a_1, \ldots, a_d) - K_{d+1}(a_0, \ldots, a_d) = -K_{d-1}(a_2, \ldots, a_d)$$

using (5), and hence

$$\frac{\partial}{\partial a_1} \frac{K_{d+1}(a_0, \ldots, a_d)}{K_d(a_1, \ldots, a_d)} = - \left( \frac{K_{d-1}(a_2, \ldots, a_d)}{K_d(a_1, \ldots, a_d)} \right)^2.$$

We generalize from the cases $k = 0$ and $k = 1$ with the following lemma.

**Lemma 3.** The following identity holds:

$$K_k(a_0, \ldots, a_{k-1}) K_d(a_1, \ldots, a_d) - K_{k-1}(a_1, \ldots, a_{k-1}) K_{d+1}(a_0, \ldots, a_d)$$
$$= (-1)^k K_{d-k}(a_{k+1}, \ldots, a_d).$$

Combining (12) and Lemma 3 completes the proof. □

*Proof of Lemma 3.* We prove the lemma by induction. The base cases $k = 0$ and $k = 1$ were shown above and hold moreover for any depth $d$ and any sequence $a_0, \ldots, a_d$. Assume then that the lemma is true for some $k$, any $d$, and any $a_0, \ldots, a_d$. For $k + 1$, we use recursion (5) to obtain

$$K_{k+1}(a_0, \ldots, a_k) K_d(a_1, \ldots, a_d) - K_k(a_1, \ldots, a_k) K_{d+1}(a_0, \ldots, a_d)$$
$$= \big( a_0 K_k(a_1, \ldots, a_k) + K_{k-1}(a_2, \ldots, a_k) \big) K_d(a_1, \ldots, a_d)$$
$$\qquad - K_k(a_1, \ldots, a_k) \big( a_0 K_d(a_1, \ldots, a_d) + K_{d-1}(a_2, \ldots, a_d) \big)$$
$$= K_{k-1}(a_2, \ldots, a_k) K_d(a_1, \ldots, a_d) - K_k(a_1, \ldots, a_k) K_{d-1}(a_2, \ldots, a_d).$$

We then recognize the last line as an instance of the identity for $k$, depth $d - 1$, and sequence $a_1, \ldots, a_d$. Applying the inductive assumption,

$$K_{k+1}(a_0, \ldots, a_k) K_d(a_1, \ldots, a_d) - K_k(a_1, \ldots, a_k) K_{d+1}(a_0, \ldots, a_d)$$
$$= -(-1)^k K_{d-1-k}(a_{k+2}, \ldots, a_d)$$
$$= (-1)^{k+1} K_{d-(k+1)}(a_{(k+1)+1}, \ldots, a_d),$$

as required. □

## C. Example Generations

In Figures 6, 7, 8 and 9 we see example generations of GPT2-xl, CoFrGeNet-F, CoFrGeNet-A and CoFrGeNet respectively when pre-trained on OWT dataset. While in Figures 10, 11, 12 and 13 we see example generations of GPT2-xl, CoFrGeNet-F, CoFrGeNet-A and CoFrGeNet respectively when pre-trained on GW dataset.

*Figure 6.* GPT2-xl example generation when pre-trained on OWT.

*Figure 7.* CoFrGeNet-F example generation when pre-trained on OWT.

*Table 9.* Downstream task accuracies on GLUE benchmark after finetuning the pre-trained models. The first column is the pre-training dataset. Results are mean±std with the best means bolded.

| Data | Model | MNLI | QQP | QNLI | SST2 | COLA | MRPC | RTE | WNLI |
|------|-------|------|-----|------|------|------|------|-----|------|
| OWT | GPT2-xl (1.5B) | 86.89±.15 | 88.93±.67 | 91.35±.34 | 93.56±.24 | 81.78±.38 | 79.83±.26 | 60.27±.22 | 58.28±.28 |
| | CoFrGeNet-F (985M) | **87.26**±.18 | **89.95**±.12 | **91.89**±.34 | **94.16**±.29 | **82.59**±.23 | **80.21**±.19 | **61.35**±.32 | **58.30**±.16 |
| | CoFrGeNet-A (1.21B) | 86.94±.12 | 89.31±.42 | 91.74±.31 | 93.83±.72 | 81.77±.25 | 79.89±.14 | 60.91±.92 | 58.28±.17 |
| | CoFrGeNet (798M) | 87.11±.09 | 89.36±.23 | 91.79±.25 | 93.91±.15 | 81.97±.14 | 79.93±.17 | 61.25±.46 | 58.29±.19 |
| | Synthesizer-D (1.2B) | 84.93±.34 | 86.82±.34 | 90.13±.51 | 91.34±.54 | 80.15±.72 | 77.95±.25 | 59.83±.35 | 58.28±.92 |
| | Sparse Attn (1.21B) | 85.27±.63 | 86.38±.33 | 90.93±.18 | 92.72±.21 | 80.76±.28 | 77.42±.41 | 59.36±.29 | 58.27±.25 |
| GW | GPT2-xl (1.5B) | 78.28±.82 | 86.83±.17 | **82.93**±.37 | 91.82±.22 | 74.18±.82 | 77.72±.93 | 60.19±.01 | **58.33**±.07 |
| | CoFrGeNet-F (985M) | **79.62**±.63 | **87.26**±.25 | 82.73±.53 | **92.36**±.45 | **74.83**±.56 | **78.01**±.34 | **61.35**±.08 | **58.33**±.04 |
| | CoFrGeNet-A (1.21B) | 78.42±.34 | 86.17±.46 | 82.51±.36 | 91.86±.36 | 74.15±.43 | 77.37±.83 | 60.85±.06 | **58.33**±.06 |
| | CoFrGeNet (798M) | 79.05±.37 | 86.98±.22 | 82.12±.28 | 92.13±.73 | 74.38±.74 | 77.95±.73 | 61.11±.04 | **58.33**±.02 |
| | Synthesizer-D (1.2B) | 77.56±.12 | 86.35±.61 | 80.38±.83 | 91.25±.71 | 73.27±.73 | 76.73±.27 | 59.26±.22 | 58.24±.97 |
| | Sparse Attn (1.21B) | 77.67±.38 | 86.41±.82 | 80.77±.16 | 91.16±.16 | 72.83±.26 | 76.62±.81 | 59.39±.38 | 58.28±.28 |

```
The court, however, found the case defies the law's prohibition that "the defendant made a false statement" and that "the defendant took an oath to the contrary, and
has disclosed this information via the internet."

Weaver's attorneys, who had filed an appeal, have pointed out that the case is part of a big-time conspiracy.

The case has been heavily criticized for alleged national security concerns. In a 1998 Washington Post interview, a federal judge ruled that the case lacked
credibility based on evidence and relied on circumstantial evidence.

"The United States' expert witness system is an embarrassment at this point," the judge wrote in the opinion. "It disregards the facts of the evidentiary evidence as
well as the evidence that led to the exclusion of any evidence of the crime."

The appeals court found that, in part, the defendant's death sentence in the original case "was not a reliable indication of the facts."

"In this light,'s judgments, which are not necessarily true, may have a negative connotation," the judge wrote, "that the jury's decision did not have the potential
to directly inform or inform a jury of their decision."

"The commission did not consider the 'reasonable doubt' that the defendant was innocent of killing, as he sought to believe, that he was unjustly accused of shooting
the suspect, likely by a mistake of his own making," the court said. "Moreover, the court's judgment that the defendant was in fact guilty of deadly force and was no
longer planning to kill anyone at the time of the shooting."

The full 12-page summary of the appellate court decision, which was subsequently enjoined, was based on a decision made by the court in June 2008 that said that the
"reasonableness" requirement of the defendant's death sentence was met.

The court said that the man had a "serious" enough mental impairment to cause him serious injury to the life of the victim in the incident."

"The circumstances of the death have not been adequately explained to prosecutors and the jury,'' the judge said in a previous order. "In essence, the plaintiff was
under the circumstances of deprivation that the allegations against him were determined by the jury and the jury."

Judge Posner ordered that the jury send to review each of the three cases in order to establish the truth of the defendant's criminal trial, which determines whether
or not Mr.
```

*Figure 8.* CoFrGeNet-A example generation when pre-trained on OWT.

```
The second reason for the need for the patent was that the court accepted the facts that the patent does not give the same benefits as the patent does. The court
established that the patent does not give the same benefits as the patent does. The court also established that the patent does not give the same benefits as the patent
does.

The third reason for the necessary appeal was that the patent did not give the same benefits as the patent does. The court established that the patent does not give the
same benefits as the patent does. The court also established that the patent does not give the same benefits as the patent does.

The fourth reason for the need for the patent was that the patent did not give the same benefits as the patent does. The court established that the patent does not give the
same benefits as the patent does. The court also established that the patent does not give the same benefits as the patent does.

The Fifth reason for the need for the patent was that the patent did not give the same benefits as the patent does. The court established that the patent does not give the
same benefits as the patent does.

The Sixth reason for the need for the patent was that the patent did not give the same benefits as the patent does. The court established that the patent does not give the
same benefits as the patent does.

The Seventh reason for the need for the patent was that the patent does not give the same benefits as the patent does. The court also established that the patent does not
give the same benefits as the patent does.

The Eighth reason for the need for the patent was that the patent does not give the same benefits as the patent does. The court also established that the patent does not
give the same benefits as the use of the software and the patent does not give the same benefits as the use of the software and the patent does not give the same benefits
as the use of the software and the patent does not give the same benefits as the use of the software and the patent does not give the same benefits as the use of the
software and the patent does not give the same benefits as the use of the software and the patent does not give the same benefits as the use of the software and the
patent does not give the same benefits as the use of the software and the patent does not give the patent does not give the patent does not give the patent does not give
the patent does not give the patent does not give the patent does
---------------
```

*Figure 9.* CoFrGeNet example generation when pre-trained on OWT.

The IANIC Cohort Study (NIH/CIRM, National Institutes of Health, and the Laboratory of Cancer Epigenetics) conducted a unique analysis of the IANIC Cohorts (ID-1058 and ID-1059), which have produced over 7,000 genetic events over the past 15 years. The IANIC Cohort study identified the most promising treatments for cancer, and focused attention on genes that improve patient outcomes. Notably, the ID-1058 study showed that the cancer-fighting properties of IANIC Cohorts are not restricted to early stages of cancer. The ID-1059 study showed that patients with the most blood cancers, including non-small tissue carcinoma, had an equivalent to 1.4 times higher disease progression than healthy patients. The Cancer Genome Atlas (CGA) identified 143 IANIC Cohorts that showed significant genomic alterations in multiple solid tumors. "The ID-1058 and ID-1059 studies strongly suggest that IANIC Cohorts may hold promise for patients with recurrent, metastatic, or relapsed brain tumors," said lead investigator Dr. Avi Mastelli, Ph.D. "But we've also shown that these findings can help identify patients at risk for developing non-small-cell lung cancer and other malignancies. These discoveries have sparked further exploration of how the body is able to defend itself against cancer."

*Figure 10.* GPT2-xl example generation when pre-trained on GneissWeb.

Sure, the more the sauce, the more they'll have to take a bit longer to get it up to the brain. But when it's consumed throughout the day, it's most likely to be absorbed into the brain long after it's already been consumed. So, if you're still trying to keep your sugar cravings at bay, use the following tips to help you stay abreast of your sugar cravings. What You Can Do If you're struggling with your cravings, here are a few actionable steps you can take to help you change your sugar cravings and keep your brain healthy:
- Limit your intake of sugar. Sugar is a great source of the sugar and can increase the amount of glucose in the blood. Limit your intake of processed or fast food items and other sources of added sugar.
- Limit your intake of sugary foods. Sugar is readily broken down, so it's best to limit your intake of sugary foods to prevent overconsumption.
- Eat properly. Sugar is broken down into glucose, which is then further broken down in the liver. Foods that have high levels of sugar are foods that are often high in fat. Once you eat foods that have high amounts of sugar, consider hydration, and other important nutrients.

*Figure 11.* CoFrGeNet-F example generation when pre-trained on GneissWeb.

While in the midst of the chaos of the pandemic, Mike is beginning to gain a greater sense of control over the reality of his job. "When I was told I was going to have to be hospitalized, my Mike was confident that he would be able to work during the pandemic," Cannon continued. "And, in the end, he found himself in the hospital. I spoke to some people who are transitioning into being a nurse to be more flexible during this time. They are a lot more relaxed than I am now."

In most cases, there are some areas of the business where the stressors of being a nurse can prove challenging. Like many, there are also quirks in the nursing profession. For example, military personnel have their own business, they may be comfortable in a hospital setting, in a commercial setting, or they may have to work long hours to maintain an idea of their job. In some cases, the pressure of being in a military setting can lead to burnout. Because of the nature of work and the nature of the jobs, there are certain occupations where they are difficult to deal with. This is particularly true for nurses, who may be feeling a greater sense of helplessness. The most common effect that nurses experience in their careers is the stress that comes with the job. In the months and years that they are in that position, they experience their own stressors. For instance, a nursing career might be quite stressful, as it can be overwhelming to create a plan or manage it. Thus, it is also important to have a plan for the future of the nurse. This can be done in many different ways depending on the person, the person's situation, and the circumstances that impact them.

*Figure 12.* CoFrGeNet-A example generation when pre-trained on GneissWeb.

A lot of these companies have been part of the product development process for decades, and that's a really good reason to be able to give it a try. I'm particularly interested in the fact that the data is open and honest, no matter what the company, the CEO, or the company. And I've seen that there's lots of open source issues, and I'm sure there's a lot of people who do the same thing.

Microsoft's open source software development program is really made up of people who share the same set of licenses, and some who think that it is not worth researching what they're requir ed to solve. Software development is an open source software company, because it's another method. We've all heard the same thing. Sometimes we get a lot of customers complaining about a particular product. We have to work on problems before they can actually address the issue. If they're applying a different company to a different product, they make their product a better product. So although FSA doesn't work, it's not a good idea to have a product. The product is a product. The second approach, the most common thing you have is the product. You have the product and the product. And finally, there usually is something that you could be doing is "I'm not doing it."

*Figure 13.* CoFrGeNet example generation when pre-trained on GneissWeb.

