# OpenReview forum: "CoFrGeNet: Continued Fraction Architectures for Language Generation"
_ICML.cc/2026/Conference — ICML 2026 regular_

### Official Review · Reviewer_XceW · 2026-02-14

**Soundness:** 4
**Presentation:** 4
**Significance:** 4
**Originality:** 4
**Overall Recommendation:** 5
**Confidence:** 1

**Summary:**

The paper introduces CoFrGeNet, a novel architecture for language generation that replaces standard Transformer Attention and FFN blocks with components based on continued fractions. The authors address the computational inefficiencies of traditional continued fractions by reformulating the gradient computation using continuants, reducing division operations to a constant complexity. Through extensive pre-training experiments on GPT-2 (1.5B) and Llama-3 (3.2B) architectures, the work demonstrates that CoFrGeNets can achieve competitive or superior performance compared to standard Transformers and efficient attention baselines, while requiring 33% to 50% fewer parameters and offering significantly faster inference speeds.

**Compliance With Llm Reviewing Policy:**

Affirmed.

**Key Questions For Authors:**

1. Regarding the CAttn mechanism, how does it handle extremely long context windows compared to mechanisms like FlashAttention?
2. Does the recursive nature of the ladder structure introduce any bottlenecks for very long sequences?

**Limitations:**

As acknowledged by the authors, the current implementation does not yet utilize custom low-level kernels (e.g., Triton), which implies the reported speedups are likely a lower bound of the architecture's true potential.

**Strengths And Weaknesses:**

# Strengths
Soundness: Excellent. The submission is technically rigorous. The authors identify a critical bottleneck in applying continued fractions to deep learning (computational cost and numerical instability of nested divisions), and solve it elegantly using the theory of continuants, enabling stable training.

Presentation: Excellent. The paper is well-structured and clearly written. The transition from the mathematical properties of continued fractions to their implementation as neural network layers (CAttn and Cffn) is intuitive. The distinction between the proposed method and prior work (CoFrNets for supervised learning) is clearly articulated.

Significance: Excellent. This work addresses the critical problem of parameter and inference efficiency in Large Language Models (LLMs). By introducing a new function class that offers a high compression ratio (up to 2x) without sacrificing generation quality, it opens a promising new direction for efficient LLM deployment. The ability to act as a "plug-and-play" replacement for Transformer blocks enhances its potential for broad adoption.

Originality: Excellent. The application of continued fractions to generative, causal modeling is highly novel. The derivation of custom gradients via continuants to bypass the auto-differentiation overhead of nested divisions is a significant theoretical contribution. This approach distinguishes itself clearly from existing sparse attention or linear attention mechanisms by fundamentally changing the underlying function approximation method.

# Weakness
Comparison with State-of-the-Art Efficient Architectures: While the comparison with standard and sparse Transformer variants is robust, the evaluation would benefit significantly from benchmarking against recent non-Transformer efficient architectures, such as Mamba (SSMs). Although the authors acknowledge Mamba in the introduction and discussion, they do not provide empirical comparisons. Given that CoFrGeNet aims to address the efficiency bottlenecks of Transformers, positioning its performance and throughput against leading linear-time sequence models would provide a more comprehensive view of its practical utility in the current landscape.

---

> ### Author Rebuttal · Authors · 2026-03-26
>
> Thank you for your thoughtful review. Below are our responses.
>
> > Mamba performance?
>
> We pre-trained a Mamba-2 3.2B model on the docling data mix. The results on the benchmarks reported in the paper are as follows:
>
> | Model | Opqa | Piqa | Arc | Wino | Hswag | Lbda | Boolq | Sciq |
> |-------|------|------|------|------|------|------|------|------|
> | Mamba-2 |  0.324 | 0.761  | 0.768 | 0.615 | 0.486 | 0.548 | 0.655 | 0.919 |
>
> When compared with Table 6 in the paper we see that the results of our variants are comparable. In fact, except Opqa on other benchmarks CoFrGeNet-F is better than Mamba-2. The throughput of Mamba-2 was roughly 300B tokens/day.
>
> > Does the recursive nature of the ladder structure introduce any bottlenecks for very long sequences?
>
> Interesting question! The ladders are made up of stacked fractions and are not recursive, and hence the representations at the output of each rung of a ladder are not overwritten, thus making it different than recursive models such as SSMs or RNNs. Because of this the signal is maintained even over long sequences.
>
> >  Regarding the CAttn mechanism, how does it handle extremely long context windows?
>
> CAttnM produces attention weights as a function of the current token and hence is very efficient in generating attention patterns even over long sequences. CAttnU looks at the entire sequence but there isn’t a quadratic matrix-matrix multiplication to produce attention weights which makes it more efficient. Moreover, both the architectures can leverage strategies already present in literature to handle long contexts such as overlapping smaller context windows or chunking or hierarchical summarization.

---

> > ### Author Rebuttal · Reviewer_XceW · 2026-04-03
> >
> > Since  author  rebuttal  solve my  concern, I keep my positive  score.

---

### Official Review · Reviewer_i184 · 2026-03-09

**Soundness:** 2
**Presentation:** 2
**Significance:** 2
**Originality:** 2
**Overall Recommendation:** 3
**Confidence:** 3

**Summary:**

This paper proposes a family of architectures based on continued fractions that serve as plug-in replacements for MHA and FF networks in modern transformer blocks. Building on prior work, the authors design causal-compatible variants using univariate ladders with upper triangular linear layers for attn replacements. They use continuant polynomials for both fwd and bwd computationm, reducing the # of div. operations from d (depth) to 1.

**Compliance With Llm Reviewing Policy:**

Affirmed.

**Final Justification:**

Rebuttal addressed some of my concerns.

**Key Questions For Authors:**

Q1 - Have you analyzed what these attention maps look like qualitatively? Do they learn meaningful token-to-token relationships, or do they converge to near-uniform/positional patterns?

**Limitations:**

Yes

**Strengths And Weaknesses:**

## Strengths

S1 - Use of polynomials to reduce div operations is pretty nice. This is a very useful and practical contribution in my view.

S2 - Parameter reductions seem significant from the reported results.

S3 - Modular design is a good experimental choice, clearly shows the benefit of FFN replacement which is useful for practical use-cases where we may want to selectively adopt one component.

## Weaknesses

W1 - Unless I am misunderstanding something very central, I don't believe CAttn attention replacement is really attention. The CAttnU variant transposes input and applies univariate ladders with upper triangular mattrices. This entirely eliminates the pairwise token-token interaction (QK^T) that defines attention in a modern transformer. I believe this is closer to a gated MLP-Mixer with a causal mask. CAttnM retains a value matrix and seems to apply a causal softmax over ladder outputs concatenated with a learned matrix. But this does **not** produce weights as a function of pairwise content similarity. The authors don't discuss this fundamental difference adequately, or analyze what expressive capacity is lost.

W2 - The improvements (table 2) seem fairly marginal. Do authors have evidence or their statistical significance? In some cases, it appears that CoFrGeNet variants even underperform GPT2-XL.

W3 - Perplexity results seem mixed and dependent on the dataset. It's hard for me to draw any conclusion re when/if CoFrGeNet is significantly better.

W4 - There's a dyadic training schedule described in the paper. Table 5 makes it seem like it's highly relevant, but doesn't this also mean that improvements may simply stem from curriculum/regularization effect of progressive depth unfreezing rather than from the continued fraction function class itself? The baselines do not receive any analogous progressive training (as best as I can tell), and that in my view makes this comparison unfair.

W5 - Generation quality evaluation: The paper is motivated as an architecture for "...language generation" but there is no systematic evaluation of generation quality. I see example generations, but no quantitative metrics and perplexity alone does not demonstrate generation quality.

---

> ### Author Rebuttal · Authors · 2026-03-26
>
> Thank you for your thoughtful review. Below are our responses.
>
> >  CAttn aren’t real attention? ... analyzed what these attention maps look like qualitatively?
>
> We view attention as a function that determines how a particular token should depend on other tokens. This function can be computed in various ways and not necessarily via pairwise similarity (dot products) of token-token representations. We believe that the performance of our models and their example generations show that our ways of computing this token-token dependence are effective. Moreover, taking your suggestion we now have **plotted** some **(causal) attention maps** (see figures in [PDF](https://anonymous.4open.science/r/cofrgenets_icml_2026_rebuttal-F5B3/attn_maps.pdf)). These maps are not simply uniform and capture meaningful relationships such as "crashed" depends on "market" in Figure 14(top), "processor" depends on "smartphone" and "powerful" in Figure 15(bottom).
>
> In CAttnU, the transpose turns sequence length into vector dimension, which when coupled with the univariate ladders and the upper triangular linear layer, ensures that a token is a function of its previous tokens. In CAttnM, the attention weights are a function of tokens, which then are used to weigh the previous tokens in the $V$ tensor. This is a linear form of attention, which through training learns meaningful attention weights (ie token-token interactions) as seen above.
>
> > The improvements (table 2) seem fairly marginal. Perplexity results seem mixed...
>
> Our intent in this work was to propose architectural options that lead to **competitive performance** with current transformer models based on a **novel function class** that requires **fewer parameters**. Surpassing performance (viz. accuracy, perplexity) is not our expectation given that we are parameter efficient. Given the novelty of the function class and architectures, we believe that even being in the ballpark in terms of performance is a meaningful contribution. The fact that we surpassed transformer models in some cases speaks to the potential of these architectures, but should not necessarily be an expectation. Please also note that the current maturity of transformer architectures (originally published in 2017) is a result of years of work by thousands of people. Re: statistical significance, Table 8 in the appendix reports standard deviations.
>
> Additionally, in a recent NeurIPS paper [1] adapting Diffusion models for language generation, the authors also pre-trained their model on OWT (and one other dataset), results were also reported on GLUE, and the size of the models was only on the 100M parameter scale. This is much smaller than what we have tested for our new architectures. Nevertheless, this paper has over 300 citations and has had significant impact.
>
> Moreover, we are seeing initial success by outsourcing divisions to FPGA -- FP32 division is 525 cycles on A100 vs 40 -- leading to potentially significant speedups.
>
> We hope you consider these aspects and the potential of these architectures as they evolve.
>
> >  improvements may simply stem from curriculum...  The baselines do not receive any analogous progressive training…
>
> Our incremental training is done *within* CoFrGeNet modules as opposed to *across* modules (the latter has been done for transformer networks [2, 3]). This incremental training is specific to the CoFrGeNet architecture: it has multiple fractions that can begin training at different times, and the main reason is to stabilize the divisions. It is not clear how a feedforward module or attention module would be incrementally trained in the same manner. Note that at the module level, we train all modules at the same time, similar to standard transformer training. Moreover, the linear part of the continued fraction is trained throughout (as it is depth 0). This linear part is the element of our architecture that is closest to a feedforward layer and is trained in the same manner. We also note that previous works [2, 3] have shown that incremental training (across modules) does not help with perplexity, etc., for standard LLM architectures. All of these facts make the comparison fair in our opinion and the architecture an integral part.
>
> > no systematic evaluation of generation quality
>
> We now have added a Table here [PDF](https://anonymous.4open.science/r/cofrgenets_icml_2026_rebuttal-F5B3/attn_maps.pdf) (see last page). We used Claude Opus 4.6 to assess the fluency, coherence and repetition rate of a sample of 100 generations for models pre-trained on OWT. The generation quality is maintained relative to GPT2-xl by our variants while requiring much fewer parameters with CoFrGeNet-F being the best.
>
> 1] S. Sahoo, et. al. Simple and effective masked diffusion language models. NeurIPS 2024.
>
> 2] M. Li, et. al. On the Effectiveness of Incremental Training of Large Language Models. arxiv 2024.
>
> 3] N. Saunshi, et. al. On the Inductive Bias of Stacking Towards Improving Reasoning. NeurIPS 2024.

---

> > ### Author Rebuttal · Reviewer_i184 · 2026-04-03
> >
> > I acknowledge author's response and have adjusted my score accordingly.

---

### Official Review · Reviewer_EhYG · 2026-03-12

**Soundness:** 3
**Presentation:** 3
**Significance:** 2
**Originality:** 3
**Overall Recommendation:** 3
**Confidence:** 3

**Summary:**

The authors propose CoFrGeNet. This architecture replaces MHA and FFN in Transformers with CoFrNets. A core theoretical contribution is the use of continuants. This mathematical reformulation allows computing continued fractions with only a single division operation. This applies to both the forward and backward passes. It solves a major computational bottleneck in prior CoFrNet architectures. The authors propose specific variants for causal attention (CAttnU, CAttnM) and FFNs (Cffn). They also introduce a dyadic training schedule to stabilize optimization. The method is evaluated by modifying GPT2-xl and Llama-3.2B architectures. Pre-training is conducted on OpenWebText, GneissWeb, and the docling data mix. Results show parameter savings and competitive performance on perplexity and GLUE benchmarks.

**Compliance With Llm Reviewing Policy:**

Affirmed.

**Key Questions For Authors:**

- Figure 8 exhibits text repetition. Could the authors quantify the repetition rate across your generated evaluations compared to standard Transformers?

- Could the authors provide FLOPs evaluations?
- Others: See weakness

**Limitations:**

yes

**Strengths And Weaknesses:**

**Strengths:**

1. The use of continuants provides a clever solution to a real computational bottleneck. Reducing divisions from $d$ to 1 is practically valuable.
2. The parameter savings are good. Table 1 demonstrates linear scaling with $p$ rather than quadratic.
3. The proposed component seems easy to plug into existing architectures.

**Weakeness:**

1. The empirical efficiency evaluation lack rigor. Table 4 compares wall-clock time. Wall-clock time is highly implementation-dependent. A rigorous comparison requires FLOP counts. (190 hours on 16 H100s for a ~1B parameter model on OpenWebText seems very slow?)
2. Can the proposed method extend to sizes like 7B/8B models, this is more practical and common? Also test more benchmarks.

---

> ### Author Rebuttal · Authors · 2026-03-26
>
> Thank you for your thoughtful review. Below are our responses.
>
> >  Could the authors provide FLOPs evaluations?
>
> Here are evaluations of training FLOPs per token for the different models corresponding to GPT2-xl and Llama 3.2B. These training FLOPs are in line with parameter counts and wall-clock times.
>
> | Model | GPT2-xl | CoFrGeNet-F | CoFrGeNet-A | CoFrGeNet |
> |-------|------|------|------|------|
> | FLOPs | 9.1GF | 6.1GF | 7.4GF  | 4.9GF |
>
> | Model | Llama | CoFrGeNet-F | CoFrGeNet-A | CoFrGeNet |
> |-------|------|------|------|------|
> | FLOPs | 19.3GF | 12.5GF | 15.1GF  | 10.7GF |
>
> > 190 hours on 16 H100s for a ~1B parameter model on OpenWebText seems very slow?
>
> We pre-trained GPT2-xl which is (slightly more than) 1.5B parameters with only DDP and hence it took 190 hours on OWT. For comparison, GPT2-small (124M) takes about 30 hours on 8 H100s (or roughly 4 days on 8 A100s see https://github.com/karpathy/nanoGPT). So given that GPT2-xl is around 12 times larger it is reasonable to expect roughly 6 times the time for double the resources.
>
> > Can the proposed method extend to sizes like 7B/8B models?
>
> Yes. Scaling is not an issue with our architecture. If anything we anticipate the gains to be greater as there is more scope for parameter efficiency. To test if the performance is similar, we started a pre-training run (after seeing your review) with Llama 7B. After training on 500B tokens on the docling data mix the validation loss for the original Llama 7B was 1.97 while for CoFrGeNet (4.1B) it was 1.98, thus supporting our expectation. We will report the final validation loss (i.e. on 2T tokens) in the paper once the run is completed.
>
> > Could the authors quantify the repetition rate across your generated evaluations compared to standard Transformers?
>
> Averaging over 100 generations on OWT using Claude Opus 4.6, the repetition rate of the different CoFrGeNet versions relative to GPT2-xl (repetition rate normalized to 1.0) was as follows: CoFrGeNet-F: 0.97, CoFrGeNet-A: 1.06 and CoFrGeNet: 1.08. Claude was instructed to measure the number of times each phrase is repeated in the text and report the average number of repetitions. Thus, the quality was pretty comparable.

---

> > ### Author Rebuttal · Reviewer_EhYG · 2026-04-03
> >
> > Thanks for the authors' rebuttal. I look forward the results on Llama 7B.

---

> > > ### Author Response · Authors · 2026-04-04
> > >
> > > Thank you for your response. Below are the results comparing Llama 7B and CoFrGeNet (4.1B) the variant where both attention and feedforward layers are replaced by our continued fraction architectures.
> > >
> > > | Model | Opqa	| Piqa | Arc | Wino | Hswag | Lbda | Boolq | Sciq |
> > > |-------|------|------|------|------|------|------|------|------|
> > > | Llama (7B) | 0.516 | 0.791 | 0.793 | 0.708 | 0.778 | 0.672 | 0.731 | 0.971 |
> > > | CoFrGeNet (4.1B) | 0.522 | 0.781 | 0.782 | 0.694 | 0.758 | 0.665 | 0.705 | 0.949 |
> > >
> > > As can be seen when compared with rows 1 and 4 in Table 6 in the paper CoFrGeNet is even closer in performance to Llama at the 7B parameter scale. The average percentage difference in performance across the 8 benchmarks at the **3.2B parameter scale was 2.275\%**, while at the **7B scale it is just 1.04\%**. This is consistent with our expectation that as scale increases there is more opportunity for compression and so the gap in performance is very likely to reduce. Hence, we anticipate that CoFrGeNet-F which was already better than Llama at the 3.2B scale on multiple benchmarks (in Table 6) should be even more performant at the 7B scale. The above results were obtained by running each model on 256 H100s using fsdp. The throughput of Llama 7B was 199B tokens/day, while that of CoFrGeNet was 281B tokens/day.

---

### Official Review · Reviewer_y3p5 · 2026-03-12

**Soundness:** 4
**Presentation:** 3
**Significance:** 4
**Originality:** 3
**Overall Recommendation:** 5
**Confidence:** 2

**Summary:**

This paper introduces CoFrGeNets (Continued Fraction Generative Networks), a novel architecture inspired by continued fractions, aimed at replacing Multi-head Attention and Feed-Forward Networks in Transformer models. The authors propose new architectural components that require fewer parameters and can be optimized more efficiently. These components are designed to be easily integrated into existing Transformer-based models, with minimal changes to training and inference procedures. The study evaluates CoFrGeNets on GPT2-xl and Llama3, showing competitive or superior performance with up to half the parameters and shorter pre-training times. The study provides a continuant based gradient derivation and a specialized training schedule to further enhance efficiency and stability. The authors believe that future optimizations tailored to hardware could further leverage the potential of CoFrGeNets, leading to more parameter-efficient and efficient generative models across different applications and industries.

**Compliance With Llm Reviewing Policy:**

Affirmed.

**Final Justification:**

The authors rebuttal addresses the concerns raised in my review. My overall assessment remains unchanged, and I maintain my positive score.

**Key Questions For Authors:**

No major questions for the authors.

**Limitations:**

Yes

**Strengths And Weaknesses:**

Strengths:

	Soundness: The paper presents a clear description of the novel CoFrGeNet architecture, providing a thorough explanation of how they can replace attention and FFNs in transformer blocks.

	Presentation: The paper is well-structured, making it easy to follow the flow of ideas from methodology to experiments. The figures effectively illustrate the proposed architectures, enhancing the reader's understanding.

	Significance: The proposed architectures offer significant parameter savings and improved efficiency, particularly in training and inference times. The experimental results demonstrate competitive performance with state-of-the-art models, indicating practical applicability.

	Originality: The introduction of CoFrGeNet architectures as alternatives to attention and FFNs addresses a critical need for more efficient and compact models. The use of continuants for efficient computation is a unique contribution that sets the work apart from existing methods.

Weaknesses:

	Soundness: While the experimental results are promising, detailed discussion on potential drawbacks or failure cases of the architecture would be helpful.

	Presentation: Some sections are math heavy, which might be challenging for readers without a strong background.

	Significance: I did not identify major weaknesses.

	Originality: I did not identify clear weaknesses regarding originality, although my familiarity with related work in this area is limited.

---

> ### Author Rebuttal · Authors · 2026-03-26
>
> Thank you for your thoughtful review. Below are our responses.
>
> > detailed discussion on potential drawbacks or failure cases of the architecture would be helpful
>
> Our use of divisions as the non-linearity distinguishes our architecture from others. However, divisions are more expensive than matrix multiplications and additions in modern digital hardware and this is a drawback. We however, are working on hardware solutions where divisions can be sent to analog hardware or FPGA and then recombined with the other computations for speed ups. We are seeing initial success in this. For instance, FP32 division on a A100 takes 525 cycles while on FPGA it takes only 40 cycles. Another solution for this issue is to rewrite the divisions in terms of logarithms and exponentials, which we are also seeing success in. The lack of an effective Triton implementation is another drawback. There also seem to be no implicit safety guards for these models, i.e., without explicit training, so similar to other architectures they may be susceptible to hallucinations, adversarial attacks and the like. We will discuss these issues in more depth in the final version.
>
> > Some sections are math heavy
>
> Given that the camera ready version will have an extra page we will describe the equations and propositions in section 4 in more detail.

---

> > ### Author Rebuttal · Reviewer_y3p5 · 2026-04-03
> >
> > I thank the authors for the rebuttal. The authors have addressed my earlier concerns. My overall assessment remains unchanged.

---

### Decision · Program_Chairs · 2026-04-30

**Decision:**

Accept (regular)

**Comment:**

This paper proposes CoFrGeNets (Continued Fraction Generative Networks), a novel architecture which is al alternative to Multi-head Attention and Feed-Forward Networks in Transformer models. The new architectural components allow competitive performance with fewer parameters. The paper derives custom gradient formulations to optimize the proposed components more accurately and efficiently than using standard PyTorch-based gradients. Experiments with GPT2-xl and Llama3 architectures show competitive or superior performance with up to half the parameters and shorter pre-training times.

Reviewers praise the originality of the proposed approach and highlight the convincing experimental results which demonstrate competitive performance with state-of-the-art models, indicating practical applicability. The main weaknesses are lack of detailed discussion on potential drawbacks or failure cases of the architecture; no comparison to non-Transformer architectures (such as Mamba or linear attention); no comparison to larger models (~7B scale); and a questioning framing of their proposed "attention" component. In the rebuttal, the authors discuss drawbacks and present convincing results on Mamba-2 and also on larger architectures (LLaMA 7B). I was less convinced about the attention component.

Overall, this is an interesting paper presenting a promising novel architecture which achieves solid results at reasonable scales. I would like to see it accepted.